# New Methods Used in Pharmacokinetics and Therapeutic Monitoring of the First and Newer Generations of Antiepileptic Drugs (AEDs)

**DOI:** 10.3390/molecules25215083

**Published:** 2020-11-02

**Authors:** Karina Sommerfeld-Klatta, Barbara Zielińska-Psuja, Marta Karaźniewcz-Łada, Franciszek K. Główka

**Affiliations:** 1Department of Toxicology, Poznan University of Medical Sciences, 60-631 Poznań, Poland; ksommerfeld@ump.edu.pl (K.S.-K.); bzielin@ump.edu.pl (B.Z.-P.); 2Department of Physical Pharmacy and Pharmacokinetics, Poznan University of Medical Sciences, 60-781 Poznań, Poland; mkaraz@ump.edu.pl

**Keywords:** antiepileptic drugs, bioanalysis, clinical samples, high-performance liquid chromatography, capillary electrophoresis, dried blood spots, pharmacokinetics, therapeutic drug monitoring

## Abstract

The review presents data from the last few years on bioanalytical methods used in therapeutic drug monitoring (TDM) of the 1st–3rd generation and the newest antiepileptic drug (AEDs) cenobamate in patients with various forms of seizures. Chemical classification, structure, mechanism of action, pharmacokinetic data and therapeutic ranges for total and free fractions and interactions were collected. The primary data on bioanalytical methods for AEDs determination included biological matrices, sample preparation, dried blood spot (DBS) analysis, column resolution, detection method, validation parameters, and clinical utility. In conclusion, the most frequently described method used in AED analysis is the LC-based technique (HPLC, UHPLC, USLC) combined with highly sensitive mass detection or fluorescence detection. However, less sensitive UV is also used. Capillary electrophoresis and gas chromatography have been rarely applied. Besides the precipitation of proteins or LLE, an automatic SPE is often a sample preparation method. Derivatization was also indicated to improve sensitivity and automate the analysis. The usefulness of the methods for TDM was also highlighted.

## 1. Introduction

About 70 million people worldwide suffer from epilepsy, a neurological disease that negatively affects patients’ quality of life and that of their families. Antiepileptic drugs (AEDs), despite many scientists’ and the medical community’s efforts, are still the basic tool in the treatment of patients with epilepsy, a disease of unknown etiology. To date, about thirty first–third generation AEDs are used in various types of epilepsy. The new AED cenobamate (CNB) was approved by the US FDA in 2019 to reduce uncontrolled partial-onset seizures in adults [1,2]. AEDs possess a long tradition in the treatment of different modes of epilepsy and over the decades many analytical methods have been developed for the therapeutic monitoring of AEDs. More improved liquid chromatography (LC) methods have also appeared in the last years, based mainly on sensitive mass spectrum detection. Generally, the elaborated methods are focused on determining many AEDs and their metabolites in one analytical run. The main problem of recent years has been developing procedures for sample micronization and automatic preparation of clinical material for quantitative determination. Microextraction techniques like solid-phase microextraction (SPME), single-drop microextraction (SDME), or dispersive liquid-liquid microextraction (DLLME) were elaborated for the isolation of many medicines, including AEDs, from biological matrices. Therapeutic Drug Monitoring (TDM) of AEDs involves measuring the drug concentrations in blood, serum, or plasma to improve the efficacy of applied treatment according to known rule *if you can’t measure it, you can’t improve it*. TDM was introduced in the late 1960s to minimize the toxicity effect caused by aminoglycosides. The main aim of TDM is to optimize drug exposure to minimize toxicities and maximize efficacy. The therapeutic tool is particularly essential when a drug that has a narrow therapeutic window and is characterized by a significant correlation between drug concentration and its toxicity, and when the clinical results depend on the drug level—total and/or free in the blood, not on the dose taken [3]. AEDs are monitored mainly due to the high inter-individual variability resulting from non-linear pharmacokinetics and the narrow therapeutic scope. New reference ranges were established for total and free concentrations of AED’s [4]. Essentials are interactions of AEDs in the pharmacokinetic phase. Carbamazepine, oxcarbazepine, vigabatrin are indicated for epileptic patients with focal seizures. Clobazam for myoclonic episodes and valproic acid are dedicated to generalized epilepsy. Levetiracetam, lamotrigine, rufinamide, topiramate, and zonisamide with a broad spectrum of action are used for most types of seizures. They require monitoring also due to the rapid absorption process, short half-life or changing with dose (nonlinearity) and significant changes in concentrations during the day or the degree of binding with blood proteins [5].

Recommendations for TDM in the treatment of epilepsy are the following [5,6,7]:small differences between the toxic effect of the drug and the symptoms of the diseasedoses of the drug that do not relieve symptomspatient belongs to one of the groups at increased risk: Elderly, children, pregnant, patients with renal and hepatic dysfunctionrequired changes in the dosage of a given drug when the patient is in weak conditionthe use of polytherapy of AED’s drugsa drug with dose-dependent—non-linear pharmacokinetics.

Currently, a great benefit for patients using various AEDs is adjusting the dosage to their individual needs and monitoring the sufficient concentrations. To correctly interpret the results obtained during TDM, pharmacokinetics knowledge on absorption, distribution, biotransformation, drug excretion and an understanding of the physiological, pathophysiological and environmental factors affecting the therapy’s success are necessary [6,8].

Initially in TDM colorimetric methods were used, and later more modern techniques were implemented, such as immunochemistry in the early 1980s, or LC. In the 2000s, the LC-MS method allowed for the quantification of free drug concentrations in biological matrices, mainly in plasma and interstitial fluids, with very sensitive detection [9]. The control of drug concentrations has provided a new perspective in the treatment in line with pharmacokinetic principles and clinical observations [7].

The matrix for TDM is often plasma or serum, rarely urine through, and increasingly unstimulated saliva, DBS or dried saliva spot (DSS). For a reliable test, the concentration of free drug in the blood has to be measured that reflects effective levels in the brain, heart tissue. In most clinical trials, the ratio between free and bounded fractions is constant at the equilibrium state. Therefore, a measurement of the total concentration (protein-bounded and free form) is acceptable in drugs with low protein binding. Such analysis is more comfortable, cheaper and requires less time expenditure [5,10].

Sample preparation is a crucial step during the whole process of the quantitative determination of the clinical samples. Simple protein precipitation or different extraction procedures before injection into LC of monitored analytes is equally important as a separation technique before detection. The most useful method for extraction includes liquid-solid, liquid-gas or liquid-liquid extraction, and mechanical separation. It is also essential to precisely determine the degree of reliability of a given method using statistical parameters known as the validation process. Validation determines the suitability of analytical methods based on the evaluation of validation parameters, like selectivity, recovery, calibration curve, determination of the linearity range, sensitivity, accuracy, precision, limit of quantification (LOQ), and limit of detection (LOD). Designed for the TDM bioanalytical method has to fulfill validation criteria established mainly by the European Medicines Agency (EMA) [11] or the Food Drug Administration (FDA) [12].

The purpose of this review is to present bioanalytical methods applied for quantification of near thirty AEDs of the 1st–3rd generation of brivaracetam (BRV), carbamazepine (CBZ), clobazam (CLB), eslicarbazepine acetate (ESL), ethosuximide (ESM), felbamate (FBM), gabapentin (GBP), lacosamide (LCM), lamotrigine (LTG), levetiracetam (LEV), perampanel (PER), phenobarbital (PHB), phenytoin (PHT), fosphenytoin (FOS), piracetam (PIR), pregabalin (PGB), primidone (PRM), oxcarbazepine (OXC), rufinamide (RFM), stiripentol (STP), sulthiame (STM), tiagabine (TGB), topiramate (TPM), valproic acid (VPA), vigabatrin (VGB) and zonisamide (ZNS) and the cenobamate (CNB), the newest AED, in biological fluids, and designed to be used in TDM in epilepsy patients. Chemical classification, structure, mechanism of action, pharmacokinetic data, and therapeutic ranges for total and free fractions and interactions of the AEDs were also collected to better view the reviewed AEDs (Figure 1 and Appendix A). Recently developed analytical methods used in therapeutic drug monitoring and pharmacokinetic studies of AEDs were presented. Details on selected biological specimens, specific extraction techniques, the column and mobile phase, the quantification level and the range of the calibration curve, and other drugs detected by the described methods have been collected in Appendix A

## 2. First-Generation Antiepileptics

### 2.1. Phenobarbital

Phenobarbital (PHB, 5-ethyl-5-phenylbarbituric acid), was introduced to treat epilepsy in 1912 and is still used worldwide. The World Health Organization has recommended the drug for treatment tonic-clonic (convulsive seizures) in developing countries [13]. The action is based on the interaction with the GABA_A_ receptor in CNS, which leads to an increase in chloride ions, and finally, neuronal excitability is reduced. PHB is extensively (>70%) metabolized by isoenzymes of cytochromes CYP2C9, CYP2C19, and CYP2E1. Conjugated metabolites are eliminated by the kidney. The elimination half-life of PHB is in wide range 30–173h, and becomes shorter with age and time of medication, because of inducing of hepatic enzymes. In children, t_0.5_ was 50 h. The renal clearance depends on pH and is increased with alkalinization of urine. In neonates and infants clearance followed 26.4 mL/h and 80.9 mL/h. Therapeutic levels are proposed to be in the range 10(15)–40 mg/L [14,15,16]. Interactions of PHB are results of inducing the effect of PHB on CYP1A2, CYP3A6, CYP2B, CYP2C, CYP3A4, and UTGs [17,18]. Inter-individual variability of the pharmacokinetics of the first-generation AEDs undergoing extensive metabolism cause the need to monitor their blood levels to obtain the desired therapeutic effect and avoid side effects. Lastly, the dispersive liquid-phase microextraction based on sequential injection solidified organic drop (DPLME-SI SFOD) coupled with the LC system, was developed for the determination of PHB and phenytoin (PHT) in urine and plasma. The potential usefulness of the method was confirmed in the analysis of real samples [19,20,21,22,23].

Especially the mode of microsampling-small volumes of blood collecting was improved. The DBS very useful for clinical applications was developed for AEDs, including PHB [20]. A new fully automated extraction system of dried blood spots, as an alternative for manual DBS extraction, coupled with LC-MS/MS was applied for the determination of the first-generation AEDs. The optimized method fulfilled validation requirements to be used for TDM. The usefulness of the developed method was confirmed on patients’ samples [21]. Besides dried blood spots, other methods of collecting blood samples have been created. Velghe and Stove applied volumetric absorptive microsampling (VAMS) for managing small volumes of blood containing PHB and other first-generation AEDs. Using VAMS influence of volumetric blood hematocrit, significant in traditional DBS, was eliminated. The extracted AEDs analytes were quantified by developed and validated UPLC-MS/MS method. Details are given in Appendix A [22]. Qu et al. [23] developed a simple bioanalytical method focused on the determination of PHB, PHT, CBZ, and its active 10,11-epoxide metabolite (CBZE) in human plasma in one analytical run. LTG was used as an internal standard. Process extraction was automated on-line solid-phase extraction (SPE); the analytical column was coupled with a high-resolution mass spectrum (HRMS). The limit of quantification (LOQ) for the PHB and PHT amounted to 0.008 mg/L and 0.0016 mg/L for CBZ and CBZE. The usefulness of the method designed for TDM was confirmed in real samples of patients with epilepsy [23]. A minimal volume of 10 µL plasma was used in the LC-MS/MS method designed for routine determination of ten AEDs using developed by Yin et al [24]. Plasma proteins were precipitated by acetonitrile. Diphenhydramine (IS-internal standard) and analytes were separated on reversed-phase C18 using the gradient flow of the mobile phase consisted of acetonitrile and water. The validated method with a limit of quantification of 0.15 mg/L for PHB and linearity of 0.01–15 mg/L was applied for TDM of over 1000 patients with epilepsy [24].

### 2.2. Phenytoin and Fosphenytoin

Phenytoin (PHT) and fosphenytoin (FOS), a water-soluble PHT prodrug used only in hospitals, are frequently used in the treatment of critically ill patients experiencing focal-onset and generalized-onset tonic-clonic seizures [25,26]. The mechanism of action is based on the modulation of voltage-gated sodium channels. The drug enhances the rapid inactivation of Na+ channels [27]. PHT is a drug whose plasma concentration is frequently monitored, yet the PHT concentration is unquestionably the most difficult to interpret pharmacokinetically. It results from a lack of predictability of the PHT plasma concentration-time profile in individuals on a given dose. The main reasons for that are high protein binding (>90%) and saturable metabolism that follows a non-linear model of Michaelis Menten pharmacokinetics [28]. Therefore, not only total but also free PHT is required to be determined to implement the appropriate dose of PHT. PHT’s therapeutic range is in range 10–20 mg/L and 1–2 mg/L for the free fraction. PHT undergoes hepatic metabolism in 98%, by the isoenzyme CYP2C9 and CYP2C9. The biological half-life of elimination is generally in the range of 7–42 h and can be extended because of non-linear pharmacokinetics. PHT is an inducer of CYP3A4, CYP2C9, CYP2C19, CYP1A2, and UGT and interacts with most antiepileptic drugs and other medications: Hormonal contraceptives, group statins, tacrolimus, ibrutinib, nilotinib, calcium channel blockers, proton-pump inhibitors [29]. To determine total PHT and other first-generation AEDs: CBZ and its 10,11-epoxide metabolite (CBZE), LC-MS/MS technique with different microextraction procedures were described in the subchapter concerning PHB [19,20,21,22,23,24]. Effortless and appropriate sensitive LC-MS/MS technique for quantifying a larger group of eight compounds of AEDs from different generations including PHT, LEV, LTG, ZNS, and TPM, divalproex, OXC, and its active metabolite was developed. Sample preparation was not complicated; 50 µL plasma spiked with IS (PHT-d10, LEV-d3, LTG-13C15N_4_, ZNS-d4, TPM-d12, OXC-d4) was precipitated using acetonitrile and diluted, then 5 µL injected to the system. The range of linearity was different for these drugs. The validated method was applied for TDM [30]. Villanelli et al. [31] developed and fully validated LC-MS/MS for quantification, only PHT in DBS. The separation was conducted on the RP column with a gradient flow mobile phase. Details are presented in Appendix A. The authors also estimated two sampling modes: Traditional in plasma and using DBS. A good correlation was found between PHT concentration determined by LC-MS/MS, both plasma and DBS sampling. The method was applied for the determination of PHT in pediatric patients and obtained results confirmed using the method for pharmacokinetic studies [31]. A free faction of PHT was also subject to bioanalysis. For determination of PHT free fraction in plasma, obtained after ultrafiltration using Millipore protein filter, validated LC-MS/MS assay was used. Linearity was in the range of 0.1–4.0 mg/L and covered a therapeutic range of free PHT. Deuterated PHT-d10 was an internal standard. Separation of the analyte was carried out on the biphenyl column with gradient flow [32].

### 2.3. Primidone

Primidone (PRM) was implemented in 1954 to treat partial-onset and generalized seizures as well as essential tremors. The mechanism of action is connected with synaptic and extrasynaptic binding for the GABA receptor. PRM after oral intake orally is absorbed in near 100%; however, smaller 60–80% is obtained, with a peak level at 3h or longer. Only 25% is bound to plasma protein. PRM undergoes hepatic metabolism by CYP2C9, CYP2C19 isoenzymes up to active metabolites: PEMA—phenylethylmalonamide and PHB. The final metabolite is responsible for the main pharmacodynamics action of PRM. PHB levels are mainly used in TDM of PRM. Renal excretion of unchanged form is in the range 40–60%. Biological half-life amounts 10–12 h for PEMA and 29–36 or longer even 75–120 h for derived PHB. Interactions similar to PHB are results of inducing effect on CYP1A2, CYP3A6, CYP2B, CYP2C, CYP3A4, and UTGs [26,29,33]. PRM and other AEDs like CBZ, PHT, PHB, and methosiximide decreased lacosamide plasma levels by 30–40% [34]. A similar interaction was observed for brivaracetam (BRV). Serum concentrations were significantly lower in patients with strong enzyme-inducing AEDs (CBZ, PHT, and/or PHB/PRM, −49%) [35]. LC-UV validated assay with on-line extraction with restricted access carbon nanotubes (RACNTs) was applied to determine PRM, PHB and CBZ. Hydantoin was used as IS. The calibration curves covered the therapeutic range for the analytes, and the method was applied for the analysis of the three AEDs in patients with mental illness [36]. A more sensitive LC-MS/MS method was used for the determination of nine AEDs, including PRM. Dried sample spot devices (DSSD) with plasma were isolated using LLE with acetonitrile and separated on the RP-18 column with gradient flow mobile phases. The validated method with high selectivity was used in TDM of over a hundred patients undergoing mono and polytherapy for epileptic diseases [37].

### 2.4. Ethosuximide

Ethosuximide (ESM) is approved for the management of absence seizures in children and adults. The medication decreases the threshold of brain currents by blocking T-type calcium channels [29]. ESM is near completely (90–100%) absorbed after oral intake and t_max_ 1–4 h and generally represents linear pharmacokinetics. However, at higher doses, clearance may be saturable (non-linear). Its protein binding is negligible (0–22%). ESM is extensively metabolized in the liver by CYP3A4 and CYP2E1 and can be induced or inhibited by other medications. T_0.5_ is 25–60 h. In children 30 h and adults 40–60 h. Excretion undergoes mainly via the kidneys and biliary [27,29,38,39].

Wu et al. [40] developed a micro-LC method with MALDI-TOF mass spectrometer to determine ESM in very low, 10 µl human plasma after prior microextraction with 40 µL volume of toluene and microderivatization to increase detection. The ESM and its derivative were separated on narrow nanoflow columns. The method was used to monitor ESM’s level after administration of 500 mg single oral tablet to healthy volunteers. Less sensitive, with LOQ 9.6 mg/L, the LC-UV method for quantification determination of ESM and seven other AEDs in clinical plasma samples was developed and fully validated by Baldelli et al. [41]. Resolution of chosen AEDs in plasma and DSSD with protein precipitation turn it out comparable. Separation of the analytes was successful using the RP-18 analytical column. The method fulfilled bioanalytical requirements to be useful in TDM of the AEDs in epilepsy patients.

### 2.5. Sulthiame

Sulthiame (STM), a cyclic sulfonamide was synthesized for the first time in the 1950s by Bayer. The mechanism of the pharmacological activity of STM is the result of inhibition of carbonic anhydrase. The medicine was effectively used for the treatment of benign epilepsy with centrotemporal spikes in 4–12 years old children. Interactions CBZ and PRM, PHT increase the elimination of STM. Antacids with magnesium and bismuth decrease gastrointestinal (GI) absorption [38,42]. Therapeutic reference range 2–10 mg/L (5–35 µmol/L) was established for STM [4]. Earlier literature data reported linear pharmacokinetics of STM. However, the most recent data point to non-linear pharmacokinetics of STM disposition (clearance was decreased with dose). A validated LC-MS/MS method was applied to determine STM in whole blood, plasma, and urine. The method with LOD of 0.01 mg/L was applied for STM pharmacokinetics in healthy volunteers [42].

## 3. Dibenzazepines

Carbamazepine (CBZ), oxcarbazepine (OXC), and eslicarbazepine acetate (ESL) are structurally related AEDs. The mechanism of action of these dibenzazepines involves inhibition of the voltage-gated sodium channel resulting in the stabilization of hyperexcited neural membranes, inhibition of repetitive neuronal firing, and diminishment of propagation of synaptic impulses. Besides, increasing potassium conductance and modulation of high voltage-activated calcium channels may also contribute to the anticonvulsant effect [43].

### 3.1. Carbamazepine

Carbamazepine (CBZ) is one of the most commonly prescribed anticonvulsants and the oldest antiepileptic drug in the dibenzazepine class. It is used as a medicine indicated for the treatment of epilepsy with partial seizures and generalized tonic seizures and mixed with the first choice of treatment. CBZ is also indicated for trigeminal neuralgia, acute maniac and mixed episodes in bipolar I disorder and schizophrenia. CBZ is highly effective and well-tolerated. The medicine is predominantly metabolized in the liver by CYP3A4 and CYP2C8 to carbamazepine-10,11-epoxide (CBZE), an active metabolite with anticonvulsant activity. Further CBZ biotransformation leads to various monohydroxylated compounds. Uridine-glucuronyl transferase UGT2B7 produces N-glucuronide of CBZ. In chronic treatment, CBZ may induce its metabolism (autoinduction) and is a strong inducer of several cytochrome P450 enzymes and UGT. CBZE is the major metabolite with >90% of total CBZ. Approximately 15% of an oral dose is excreted in the urine as N-glucuronide, and less than 2% is excreted unchanged. Non-linear pharmacokinetics of CBZ (due to autoinduction), narrow therapeutic index and unpredictable relationship between its dose and plasma levels in relation to genetics, age, sex, autoinduction, etc. make TDM useful tool in CBZ therapy. In practice, both CBZ and CBZE are monitored. The current reference range for CBZ and CBZE in human plasma is 4–12 mg/L and up to 2.3 mg/L, respectively [10,29,44,45,46].

There are different methods (GC, LC, HPLC, High Performance Thin Layer Chromatography (HPTLC), LC-MS/MS, UHPLC-MS/MS, SFC-ESI-MS/MS) to measure the level of CBZ and its major metabolite in biological samples of epilepsy patients treated with this AED in monotherapy or polytherapy. The presented methods use biological samples such as: Plasma [47,48,49,50,51,52,53,54,55], serum [53,56,57,58], urine [47,59], saliva [60], dried saliva [61] and respiratory condensate [62,63], dried blood spot [20,64,65]. In TDM of CBZ, plasma, serum, and urine are the most commonly used. In the dried blood spot method, Whatman 903 Protein Saver Cards and the LC with triple quadrupole mass chromatography were used [64]. Saliva is particularly useful in children, and the correlation between blood and saliva CBZ and its metabolite concentration was confirmed [60]. The described methods differ in the sample preparation technique including liquid-liquid extraction (LLE) [53,55,56,61], ultrasound-assisted emulsification microextraction (SAEME) [47,50], microextraction by packed sorbent (MEPS) [55,66] and deproteinization [48,49,57,58]. The most frequently used analytical method is RP-HPLC with a C18 column. The fluorimetric method is used to directly analyze the concentration of CBZ in the exhaled air condensate (EBC) [62]. HPLC-MS/MS methods are used to determine the parent AED and its metabolite [60] or parent drug simultaneously with other antiepileptic drugs. They can be used in patients treated with AEDs polytherapy [53,55,58,67]. HPLC-MS/MS, compared to other techniques, is an important tool in TDM and clinical laboratory routine due to its high sensitivity and specificity. The mobile phase is usually polar and simple and consists of a mixture of water and an organic solvent (methanol, ethanol, 2-propanol, isopropyl alcohol, *n*-hexane or acetonitrile) [48,49,50,52,53,57,58,61,66]. In some methods, acid modifiers (acetic acid, formic acid, phosphoric acid,) are added [50,57,58]. The elution system usually is isocratic. Detection is achieved by the use of spectrophotometric methods (UV) with various wavelengths (between 210 nm and 285 nm). In MS-MS detection with selective, both positive or negative electrospray ionization is also used, and analytes are measured in the multiple reaction monitoring modes (MRM) with a specific mass transition [54,68]. The described methods are selective, work in therapeutic range and are suitable for TDM of CBZ [69].

### 3.2. Oxcarbazepine

Oxcarbazepine (OXC) is an antiepileptic drug with a similar chemical structure to carbamazepine, but with different metabolism. OXC is the first-line drug for the treatment of generalized tonic-clonic and partial seizure. It is indicated in monotherapy or adjunctive therapy in adults and children of 6 years of age and above [70]. OXC is a prodrug and after oral administration is rapidly reduced in the liver, by cytosolic aryl ketone reductase to the racemic 10–hydroxycarbazepine (MHD) ((*S*)-licarbazepine, eslicarbazepine (80%), and (*R*)-licarbazepine (20%)). MHD is metabolized primarily by glucuronidation, and it also undergoes hydroxylation via CYP izoenzymes. The pharmacological activity of OXC is mainly exerted through the metabolite MHD. OXC and MHD inhibit CYP2C19 and induce CYP3A4/5, which promotes metabolic interactions with other drugs [29,71,72]. OXC metabolism is influenced by other AEDs as CBZ, PHP, PHT. In routine practice, only 10-hydroxycarbazepine is monitored and recommended for safe and effective use of OXC. MHD pharmacokinetics varied considerably among individuals [10]. The current reference for MHD in plasma is 3–35 mg/L [29].

The analytes were determined in biological matrices such as: Plasma [55,66,67,73,74,75,76,77], serum [78], urine [79], dried blood spot [37]. The methods differ in the sample preparation technique and include liquid-liquid extraction (LLE) [55], salting assisted liquid-liquid extraction (SALLE) [50], SAEME, SPE [73,75,76], micro-extraction by MEPS [66] and deproteinization [67,73,74,77,78].

The use of a SALLE technique increases the ionic strength of the entire solution. It reduces the solubility of the simultaneously determined drugs, facilitating their extraction and eliminating the drying step [50]. Analytical methods reported for the analysis of OXC include, HPLC with UV or DAD detection, and reverse elution on C18 chromatographic column [66,73,74,75]. The LC, HPLC and HPTLC methods often are coupled with a triple quadrupole mass spectrometer operated in electrospray ionization [67,76,77,78]. The most frequently used analytical method is HPLC with isocratic elution of mobile phase, and UV or DAD detection [66,73,74,75]. LC-MS/MS, HPLC-MS/MS and UHPLC-MS/MS methods are used to determine the parent OXC and its metabolite MHD [67], or parent drug simultaneously with other antiepileptic drugs [76,77,78]. SFC-ESI-MS/MS (supercritical fluid chromatography/mass spectrometry) is used to determining of OXC and MHD and co-administered AEDs (CBZ and TPM) after LLE with ethyl acetone [55]. The mobile phase usually consists of water and an organic solvent (methanol, carbon dioxide, ammonium acetate, *n*-hexane, isopropyl alcohol), and sometimes acid modifiers (acetic acid, formic acid) are added [67,77,78]. The elution system usually is isocratic. Detection is achieved by UV methods of various wavelengths. A triple quadrupole with electrospray ionization is used for MS-MS detection. Analytes are measured in the multiple reaction monitoring modes with a specific mass transition [55,67,76,77,78]. The described methods for determining OXC in epilepsy patients are selective, operate in the therapeutic range and are suitable for both TDM and pharmacokinetic studies.

### 3.3. Eslicarbazepine Acetate

Eslicarbazepine acetate (ESL), is the third generation of the carboxamide family (CBZ, OXC). It is indicated as either adjunctive or monotherapy treatment for partial seizures in adult and children four years of age and older [80,81]. The main advantages of ESL are its administration in a single daily dose, the lack of significant interactions with other drugs and a favorable safety profile. As a prodrug, it is almost immediately metabolized to its enantiomeric monohydroxy derivatives—ESL; (*S*)-licarbazepine (80%) and (*R*)-licarbazepine (20%). ESL, the anticonvulsant metabolite, is subsequently metabolized by conjugation with glucuronic acid via UGT isozymes (1A4, 1A9, 2B4, 2B7, 2B17) [82]. ESL has the same mode of action as OXC and is most closely related to CBZ. It acts as a competitive blocker of voltage-gated sodium channels and stabilizes their inactive state. ESL, compared to CBZ and OXC, has a lower affinity for inactive channels and limits their availability by selectively increasing slow inactivation. ESL, like OXC, is a weak inducer of the enzyme CYP3A4 and UDP-glucuronosyltransferase, therefore, undergoes very little pharmacokinetic drug-drug interaction. However, ESL may reduce the effectiveness of the oral contraceptive hormone by induction of CYP3A4 and may affect PHT metabolism by inhibition of CYP2C9 and CYP2C19. Conversely, CBZ, PHT, PB and TPM increase the elimination of ESL and lower its plasma concentration [10,29,45,83]. There is a linear relationship between the dose of ESL and serum concentration at clinically relevant doses of 400–1600 mg/day [45,82]. The current reference range for ESL in human plasma is 3–26 mg/L [4]. Plasma or serum samples are generally used for TDM of ESL [50,53,75,76]. There are only a few different representative analytical methods used to determine ESL in biological samples, including HPTLC [50], UHPLC-MS/MS [53], LC-MS/MS [84]. The most commonly used are a C18 column and isocratic elution. For sample preparation, techniques such as SALLE [53], SPE [50], one step extraction combined with the precipitation of proteins and phospholipids [53] and protein precipitation [84] are most used.

### 3.4. Clobazam

Clobazam (CLB) is a 1,5-benzodiazepine and partial γ-aminobutyric acid (GABA) receptor agonist, with anxiolytic, sedative, and anticonvulsant activities. CLB binds to stereospecific benzodiazepine receptors on the postsynaptic GABA neuron at several sites in the central nervous system. CLB is highly selective for the α2 subunit of the GABA A receptor-associated with antiepileptic activity and less selective for the α1 subunit. After oral ingestion, CLB is rapidly and completely absorbed (100%) [10,29,85]. The CLB pharmacokinetics is linear [85]. CLB is metabolized in the liver by P450 isoenzymes to an active metabolite N-desmethylclobazam (N-CLB) and other metabolites (over 20). The cytochrome P450 CYP3A4 is responsible for the conversion of CLB to N-CLB, which is further biotransformed by CYP2C19 to 4′-hydroxy-N-CLB. Plasma concentrations of N-CLB are five times higher in CYP2C19 poor metabolizers versus extensive metabolizers. Binding of CLB and N-CLB to plasma proteins is approximately 90%. CLB is eliminated via the urine (~94%) as metabolites. CYP-dependent metabolism of CLB may be subject to drug-drug interactions [10,85]. CYP3A inducing antiepileptic drugs (PB, PHT, CBZ) do not affect CLB levels but increase N-CLB levels. CYP2C19 inducers may increase N-CLB excretion, while inhibitors do not affect N-CLB elimination. FBM increases the concentration of N-CLB in the serum. STP is an inhibitor of CLB and N-CLB biotransformation and increases the concentration of N-CLB several times. In patients treated with CLB in therapeutic doses, the current plasma reference range for parent drug is 0.03–0.30 mg/L and 0.3–3.0 mg/L for N-CLB [10,29]. There are a few bioanalytical methods used for quantitative estimation of CLB and its N-CLB metabolite. In LC-MS/MS methods, the purification sample step includes LLE with ethyl acetate and protein precipitation with methanol. The C18 columns with isocratic mobile phase are used [86]. The detection is performed by LC coupled with triple quadrupole mass spectrometer operated in MRM mode and selected reaction monitoring (SRM) after LLE with ethyl acetate and reconstitution by acetonitrile:water or protein precipitation with methanol [87,88]. DBS, as an alternative to conventional plasma sample, are used for simultaneous quantitation of 15 benzodiazepines. The whole blood was spotted on a filter paper card protein saver 903 and LC-MS/MS analysis was performed [86]. DBS sampling seems to be a promising technique in the TDM of CLB.

## 4. Piracetam and Its Newer Derivatives

### 4.1. Piracetam

Piracetam (PIR) was introduced in the late 1960s as a promising nootropic modulator of cerebral function in patients with various encephalopathies like cognitive disorders, dementia, vertigo, cortical myoclonus, dyslexia, and sickle cell anemia. As an anticonvulsant, it is used mainly for myoclonus. PIR is not approved by the FDA. Very popular in Europe for its neuroprotective action, it has positive effects on the vascular system by reducing erythrocyte adhesion to vascular endothelium, and facilitate microcirculation without causing sedation or stimulation. PIR modulates the cholinergic, serotonergic, noradrenergic, and glutamatergic neurotransmission, but its pharmacology is not well known until today. Its mechanism includes AMPA-type glutamate receptors. PIR, with its linear and time-dependent pharmacokinetics, has low individual variability even though the doses are high (the standard dosage of PIR for children during ADHD treatment is between 40–100 mg/kg b.w. and for adults, the recommended dose is 1000–1200 mg). PIR has almost absolute bioavailability in the oral form (100%), and its half-life is about 5 h. Therefore, it should be used in divided doses (2–3 times a day). PIR absorbed following oral administration with the peak plasma concentration reached within 1 h after dosing, crosses the blood-brain barrier, with no protein binding, is excreted as unchanged drug with no metabolites. The therapeutic range is unknown. Therefore, the cases of overdose are sporadic. PIR has no severe or profound interactions with other medications. Moderate interactions are observed with clopidogrel and mild with levothyroxine. The daily dose must be individualized according to renal function, especially in elderly patients [89]. Detection methods are not very common in clinical aspects.

In most cases, the available methodological data come from the 1990s and early 2000s. There are some results of the therapeutic control of piracetam blood level in parturients and newborn infants presented by Sirotina et al., in Russia [90]. The first instrumental methods for PIR analysis in plasma, serum, urine or cerebrospinal fluid were published many years ago. They included HPLC or micellar electrokinetic chromatography (MEKC) techniques, both with UV detection [91,92,93]. There are no new analytical solutions to detect and monitor this drug. However, all available methods apply to analyze piracetam in patients with aphasia or to complete the knowledge of pharmacokinetics studies after its administration. The linear ranges are between 1–500 mg/L in blood and 100–2000 mg/L in urine, with the detection limit mostly on the level of 1 mg/L. The latest analytical reports do not directly concern TDM. They are focused on the assessment of pharmacokinetic interaction between PIR and l-carnitine in human plasma using piracetam d-8 as the internal standard and one-step precipitation of protein using acetonitrile. The extracts were analyzed by HPLC-MS/MS [94]. The HPLC methods describe using a mixture of an aqueous solution of perchloric acid, methanol or acetonitrile with protein precipitation and LLE with chloroform, 2-propanol, toluene or hexane mixture as a form of simple extraction [95,96].

### 4.2. Levetiracetam and Brivaracetam

New piracetam derivatives have been found on the drug market for several decades. Levetiracetam (LEV) and brivaracetam (BRV) are characterized by a different mechanism than PIR, including binding at the synaptic vesicle protein 2A called SV2A on its receptors site, represent a unique form of action on the newer generation of epilepsy therapy [97]. LEV was approved as the first new derivative of piracetam in 1999, and BRV seventeen years later. Both are recommended in focal and generalized epilepsies. The efficacy of LEV and BRV was proven in patients with photosensitive and difficult-to-treat focal epilepsy. In the comparison between LEV and BRV, it has been shown that LEV may be slightly more effective and less likely to feel dizzy, but both of them tend to offer better tolerance to psychiatric adverse events than older AEDs. The pharmacokinetics of LEV and BRV are very similar, with rapid and almost complete absorption. The time of peak serum level occurs 1 h for BRV, and 1.3 h for LEV. Both show fast passage across the blood-brain barrier. LEV is not bound to plasma protein, and it is metabolized to an inactive metabolite called L057. BRV’s protein binding amounts 20%, and its metabolism depends on cytochrome P450 (CYP2C9) [98,99]. There is no need to reduce the LEV dose (start in focal seizures with 250–500 mg) in hepatic impairment unless there is renal dysfunction. Nearly total excretion by the kidneys is typical for LEV and BRV, 66% of LEV and less than 10% of BRV is eliminated as unchanged compound. TDM of LEV and BRV is not routinely recommended [100,101,102]. However, dosing of both drugs is complicated in polytherapy, kidney failure or pregnancy. Therefore, their monitoring in the blood is required. For this purpose, chromatographic methods, including HPLC or GC, combined with MS or nitrogen-phosphorus detector (NPD) were applied [103,104,105]. A sensitive UHPLC-MS/MS assay was developed for simultaneous determination of BRV and its metabolites in plasma during TDM in epilepsy patients [106]. Several years ago, LC-MS/MS was developed that allows the rapid quantification of LEV and its carboxylic metabolite in human plasma [104]. Simple UHPLC method with DAD detection was validated for determining LEV without any interference from complex matrices like urine [107]. The spectrofluorimetric method has been developed for LEV determination in plasma using derivatization procedure [105]. An LC-MS/MS method for simultaneous determination LEV and 14 antipsychotics in hair was developed. The method was applied for monitoring the inappropriate use of antipsychotics in suspected of posing and in mental health patients [108]. The HPLC-DAD technique for determination of LEV in human plasma with LCM and ZNS was developed. The method was applied for TDM of the drugs [109]. Commonly used mobile phases are composed of formic acid in water with acetonitrile for BRV and phosphate buffer/methanol for LEV [110,111]. Sample preparation included protein precipitation with methanol or acetonitrile and SPE offline [103,111,112,113,114]. Less popular is LLE with dichloromethane, methanol or tert-butyl methyl ether [106,115]. Limits of the quantitation of these methods range from 0.001 mg/L to 10 mg/L for BRV (therapeutic range 0.2–2 mg/L) and 0.03 to 100 mg/L for LEV (5–41 mg/L) [103,106]. Immunoassay techniques, including enzyme-multiplied immunoassay technique (EMIT), were applied for the quantitative determination of LEV in human serum or plasma on automated clinical chemistry analyzers [116].

## 5. Topiramate

Topiramate (TPM) is a medicine with an established position among the second-generation AEDs approved for treatment focal and primary generalized-onset tonic-clonic seizures, including Lennox-Gastaut Syndrome. It is also used to treat bipolar disorder, post-traumatic stress, mood instability disorder, bulimia nervosa, weight loss in obese, and prevent migraine headaches. Multiple action mechanisms involve inhibition of voltage-dependent sodium and calcium channels; it also inhibits carbonic anhydrase activity and enhances the inhibitory effect of GABA [26]. In monotherapy, 20–30% of TPM is metabolized. In polytherapy, increases up to 50–70%, when administered together with CBZ and PHT, the clearance is increased 2-fold. Its biological half-life amounts to 20–30 h and becomes up to 50% shorter after administration with enzyme-inducing AEDs. TPM undergoes interactions not only with some AEDs but also with other groups. Over 200 drugs are known to interact with TPM [45,117]. New therapeutic reference range of 2–10 mg/L (5–35 µmol/L) was proposed for TPM, which is smaller than the previous, 5–20 mg/L [4].

Bioanalytical methods based on LC-MS/MS techniques were developed to determine TPM as a single analyte or a mixture of other AEDs. Also, different detectors were used for the purpose. Milosheska et al. [118] developed a simple HPLC method with a sensitive fluorescence detector for determination TPM in plasma. The technique required precolumn derivatization using 4-chlor-7-nitrobenzo-furazan and bendroflumethiazide as IS. Traditional LLE with ethyl acetate and diethyl ether was applied form isolation of the analyte from plasma. The validated method has been used to determine TPM in plasma patients with epilepsy [118]. To determine TPM and its four metabolites in plasma, Milosheska et al. [119], developed UPLC-MS/MS with linearity 0.1–20 mg/L using only 100 µL aliquot plasma. If compare to LC-FLD [118] with linearity 0.01–24 mg/L and using 500 µL sample, the last method looks to be more sensitive. However, the technique required a derivatization procedure and, without doubts, consumes more time and solvents. In both modes, the LLE extraction method was used. Hahn et al. [120] applied the GC-MS technique for quantification of TPM in capillary blood using DBS. The analyte and IS were isolated using the LLE method with tris buffer pH 9.5 and ethyl acetate. The method’s usefulness was confirmed by the determination of TPM in DBS obtained from phlebotomy and finger pricks of adult volunteers after taking a single oral dose of 100 mg. TPM’s higher concentrations were determined in DBS with capillary blood, and DBS obtained after phlebotomy in volunteers than in plasma [120]. Ni et al. [121] developed sensitive the LC-MS/MS method with LOQ 1 ng/mL, with positive/negative ESI. The method was applied for the pharmacokinetic study of TPM and phentermine as a combination approved by the FDA to treat obesity. The analytes were determined in plasma samples after administering extended-release capsules to male volunteers, during one analytical run [121]. Ishikawa et al. [122] proposed use CE with capacitively coupled contactless conductivity detection (C4-D) for the determination of TPM in a small volume of plasma. A validated method with confirmed high selectivity, precision, and accuracy was applied for quantification of TPM in samples obtained from patients treated with TPM.

## 6. Felbamate

Felbamate (FBM) is called a broad-spectrum AED because of the high risk of hepatotoxicity and aplastic anemia. It was approved in 1993 to treat partial seizures in adults and Lennox-Gastaut Syndrome in children [70]. FBM is a weak inhibitor on GABA and benzodiazepine receptor binding sites. With good bioavailability (>90%), a half-life of 20–23 h and low protein binding (20–36%), FBM is excreted as unchanged drug in 50% of the dose. Its therapeutic range is 30–60 mg/L. FBM interactions with inducers of liver metabolism like CBZ, PHT or PHB are very strong. Therefore, there are appropriate indications for FBM therapy blood monitoring because of variable metabolism and differences between children and adults in clearance [123,124,125]. In most cases, the available methodological data come from the 1990s and early 2000s. Many instrumental methods are used for felbamate TDM in serum, plasma and dried plasma spot (DPS) like HPLC or GC methods, with UV, nitrogen-phosphorus detector (NPD) or MS detection [41,68,126,127]. The linear ranges are between 5–105 mg/L in plasma and 2.4–96 mg/L in DPS, with the detection limit of the drug mostly on the level of 4–5 mg/L. The extraction technique generally presents LLE with methylene chloride or protein precipitation with acetonitrile or methanol. There is also a SPE procedure used in GC-NPD analysis developed by Gur et al. [126]. Shibata et al. [68] determined 22 antiepileptic drugs by UHPLC-MS/MS with a good correlation between the concentration of clinical samples and the conventional techniques like PETINIA (automated immunoassay) or simple HPLC method.

## 7. Valproic Acid

Valproic acid (VPA) is used primarily in the treatment of epilepsy and seizures, but also in migraine, bipolar, mood, anxiety, and psychiatric disorders. The drug increases the inhibitory activity of GABA through several mechanisms, including inhibition of GABA degradation, increased GABA synthesis, and decreased turnover. Moreover, VPA blocks voltage-gated ion channels and also acts as a histone deacetylase (HDAC) inhibitor [128].

VPA is almost completely metabolized in the liver by glucuronidation (50%), β -oxidation (40%), and cytochrome P450-mediated oxidation (10%) to 2-ene-VPA and hepatotoxic 4-ene-VPA [129]. Due to large interindividual differences in rate of drug metabolism, there is an unpredictable relationship between VPA dose and serum concentration. This effect is observed mainly in patients co-treated with other antiepileptics. VPA concentrations may be increased when co-administered with FBM, CLB and STP. TPM, ESM and methsuximide can decrease the drug levels [130,131].

The low correlation between VPA dose and concentration requires the individualization of therapy using TDM. Moreover, measurement of non-protein-bound concentration may be more useful clinically because of the large variability in protein binding. Samples for measurements should be taken before the morning dose [10].

There are several methods for the analysis of VPA and its metabolites in human serum, plasma and saliva. GC methods using capillary columns and nitrogen or helium as a carrier gas are available [132,133,134]. These methods use a flame ionization detector (FID) for detection of the analyte [133,134] and its metabolite 3-heptanone [132]. VPA concentrations can be determined by HPLC methods with UV detection [135,136]. Recently, LC methods coupled with MS/MS detection in the negative ion mode were reported allowing to determine VPA and its metabolites 2-ene-VPA, 4-ene-VPA and 2,4-diene-VPA in plasma and serum [137,138]. The analytes are determined using a mixture of an aqueous mobile phase with an organic solvent such as methanol or acetonitrile. Commonly used aqueous components of mobile phases are phosphate buffers of pH 3–3.5 [135,136] and ammonium acetate solution [138]. For preconditioning of samples before chromatographic analysis of VPA and its metabolites, protein precipitation followed by an extraction step is usually applied. Protein precipitation is performed using trifluoroacetic acid or acetonitrile [132,133]. For further sample purification, LLE is additionally applied using chloroform [134] and n-hexane [135]. Liquid-liquid microextraction (LLME) with chloroform [132,133] and SPE techniques [137,138] are used as well. LOQ of these methods ranging from 0.2 mg/L [132] to 25 mg/L [134] are sufficient for analysis of VPA within the therapeutic range of 50–100 mg/L with appropriate precision and accuracy. Moreover, various commercial reagent-based immunoassay techniques are available, including chemiluminescent microparticle immunoassay (CMIA) and fluorescence polarization immunoassays (FPIA) [139,140]. However, they can be subject to interferences and cross-reactivity with similar compounds.

## 8. GABA Analogues

### 8.1. Gabapentin, Pregabalin and Tiagabine

GABA analogues including gabapentin (GBP), pregabalin (PGB) and tiagabine (TGB) are structurally related to the neurotransmitter GABA, but they differ in mechanism of action. GBP and PGB inhibit the alpha 2-delta subunit of voltage-gated calcium channels, TGB inhibits GABA reuptake into neurons and glia [141,142]. GBP and PGB are approved for the management of neuropathic pain and as adjunctive therapy of partial seizures. They are used off-label for the treatment of various anxiety disorders [143,144]. Pharmacokinetics of PGB is linear, while GBP exhibits saturable absorption after oral administration leading to a non-linear increase of plasma concentration at higher doses. Both drugs undergo little or no metabolism and are excreted unchanged by the kidney [145]. Due to the dose-dependent bioavailability and interindividual variation in the pharmacokinetics of GBP, TDM can be beneficial in patients using the drug. Samples for measurements should be drawn before the morning doses [10]. Recently, for quantification of GBP in serum and blood, sensitive and validated LC procedures with MS/MS and TOF-MS detection were reported [78,146,147]. Chromatographic separation was performed on C18 columns using compositions of methanol and acetonitrile with water or ammonium acetate solution as a mobile phase (Appendix A). GC-MS methods were described for the determination of GBP in human serum after derivatization with hexyl chloroformate [148], and in DBS after microwave-assisted derivatization with heptafluorobutanol [149]. The latter study demonstrated that DBS are a valid alternative to serum for the determination of GBP. The described assays were proved to be appropriate also for the analysis of PGB in serum and blood [146,147,148]. Moreover, the validated methods for determination of the drug in plasma were reported, including LC-FLD method after derivatization with 4-fluoro-7-nitrobenzofurazan [150] and highly sensitive LC-MS/MS method [151].

Isolation of both drugs from biological matrix before chromatographic analysis was performed using protein precipitation with acetonitrile [78,147] and methanol [146], or SPE using cation exchange cartridges [151]. The LC-MS/MS methods were also reported for determination of PGB in DBS/DPS [151,152]. The drug was isolated from the matrix using LLE with a mixture of methyl tert-butyl ether and diethyl ether [151] or ethyl acetate [152].

TGB is useful in the control of partial seizures [153]. It is extensively metabolized by CYP3A to at least four pharmacologically inactive metabolites. The plasma clearance can be increased in patients treated with the enzyme-inducing antiepileptic drugs such as CBZ, PHT or PRM. Conversely, reduced clearance is observed in hepatic impairment [130]. The application of TDM for TGB is complicated because of very low concentrations of the drug in the nanomolar range (or ng/L), the short elimination half-life of 3.8–8.1 h and significant fluctuation of concentrations between dosing intervals (Appendix A). As a result, methods for the determination of the drug in the human body were not published recently. The latest one is the LC-MS/MS analysis of TGB and 21 other antiepileptics in postmortem blood, serum and plasma. The LOQ of the method for TGB is 0.05 ng/mL, which allows determining the very low concentration of the drug [84].

### 8.2. Vigabatrin

Vigabatrin (VGB) is used as adjunctive therapy in resistant epilepsy, and for monotherapy in infantile spasms in West Syndrome. The drug increases brain GABA levels through selective and irreversible inhibition of GABA-transaminase, the enzyme responsible for the degradation of GABA in the central nervous system [141,154]. VGB is highly soluble in water, does not bind to serum proteins and is not metabolized by the liver. It is marketed as a racemic mixture of two enantiomers, the pharmacologically active (+)-*S*-enantiomer, and the inactive (−)-*R*-enantiomer [155]. Due to the differences in pharmacodynamic and pharmacokinetic properties of the two enantiomers, an enantioselective method should be applied for the determination of the (+)-*S* and (−)-*R* VGB. There is a non-enantioselective GC-MS method for the determination of VGB in the presence of other antiepileptics in serum [148]. Moreover, LC-FLD method was reported for analysis of the drug in the presence of GABA and taurine in biological samples, including human plasma, rat plasma, brain and retina. The sample pre-treatment involved plasma protein precipitation with acetonitrile, and further derivatization of the analytes with naphthalene 2,3‑dicarboxaldehyde to produce stable fluorescent active isoindoles. The method was applied for the determination of VGB in rat plasma, brain and retina after intraperitoneal administration of the drug. However, validation parameters of the method were presented not only for the rat biological fluids but also for human plasma [156]. An enantioselective UHPLC-MS/MS method for determination of (−)-*R* and (+)-*S* VGB in human plasma was reported by Duhamel et al. [157]. The method involved pre-column derivatization of the enantiomers using o-phthaldialdehyde and N-acetyl-L-cysteine. Resulting diastereomeric isoindole derivatives were separated on a C18 column and were determined with LOQ of 0.2 mg/L. The method was applied for pharmacokinetic studies of VGB enantiomers in children with West Syndrome.

## 9. Lamotrigine

Lamotrigine (LTG) is a second-generation antiepileptic used for focal and generalized seizures in adults and children, and as a first-line drug for the treatment of pregnant women. It inhibits voltage-sensitive sodium channels and modulates the release of excitatory neurotransmitters such as aspartate and glutamate [158]. The drug exhibits first-order linear pharmacokinetics at therapeutic dosage [159]. It undergoes extensive metabolism to an inactive glucuronide metabolite. Enzyme-inducing anti-epileptics reduces the LTG half-life, while VPA inhibits LTG metabolism resulting in raised serum levels [158]. Interindividual variation in serum concentrations due to drug interactions and changes in drug concentration during pregnancy suggest that the application of TDM may facilitate the effectiveness and safety of the therapy with LTG. Many methods have been described for the determination of the drug in plasma and saliva, and they are based on LC with UV detection [74,160,161,162,163]. In the reported assays, the chromatographic separation was performed on a C18 column using a mobile phase composed of acetonitrile or methanol with phosphate buffer or trifluoroacetate. For isolation of the drug from a biological matrix, simple protein precipitation with acetonitrile, methanol or 10% acetic acid was applied (Appendix A). Moreover, LLE with diethyl ether [163] or its mixture with dichloromethane [160] was also reported.

The validated LC-MS/MS methods for the analysis of LTG in plasma, serum and DPS in the presence of other antiepileptics were also available [30,37,77,78,164]. The assays were characterized by high sensitivity with LOQ for LTG ranged from 0.005 [30] to 0.1 mg/L [78] in plasma and 0.25 mg/L in DPS [37]. Domingues et al. [165] described an LC-MS/MS method for determination of LTG in plasma in the presence of numerous antipsychotics, antidepressants and anxiolytics. The method was successfully applied for TDM in schizophrenic patients.

Moreover, a homogenous particle-enhanced turbidimetric immunoassay is available, which showed relatively good agreement when compared to the validated LC-UV method (R = 0.968). The linearity of measurements for the immunoassay was confirmed in the range of 2–40 mg/L [161].

## 10. Zonisamide

Zonisamide (ZNS) is a benzisoxazole derivative and has a unique structure unrelated to that of any other AED. ZNS is used as an adjunct treatment for partial seizure epilepsy and off label for bipolar disorder, chronic pain, migraine and myoclonic dystonia [29,45,166]. The exact mechanism of the anticonvulsant activity of ZNS is unknown, but current evidence suggests that the drug exerts its pharmacological effects by blockade of neuronal voltage-dependent sodium and T-type calcium channels, thereby suppressing neuronal hyper-synchronization. ZNS modulates GABA-ergic and glutamatergic neurotransmission and is a weak inhibitor of carbonic anhydrase. It is rapidly absorbed after oral ingestion. ZNS demonstrates extensive oral bioavailability (≥90%) and dose-dependent pharmacokinetics. Plasma protein binding is modest (40–50%) and ZNS preferentially accumulates in the red blood cells. The drug exhibits linear pharmacokinetics with metabolism via phase I and phase II biotransformation pathways [10,29,45,166]. The metabolism undergoes by the CYP3A4 isoenzyme (70%), but also by CYP3A7 and CYP3A5, to 2-(sulphamoylacetyl)phenol. Metabolites are not biologically active, and 35% of the drug is excreted in the urine. CBZ, PHT, PRM and risperidone enhance the elimination of ZNS and decrease its plasma concentration [166]. The current reference range for ZNS in plasma is 10–40 mg/L [10,29]. Several techniques have been developed to quantify ZNS in human matrices including HPLC or UHPLC methods with UV detection at various wavelengths in the 235–312 nm range [51,167,168] and with MS/MS detection [68,76,78]. The techniques used to prepare the sample before analysis include the precipitation of protein (PP) with acetonitrile, trichloroacetic acid or methanol [51,68,78,109,167], LLE with ethyl acetate [168], surfactant-assisted liquid-liquid dispersion microextraction (SA-DLLME) [51], MEPS [167] or reaction with NBD-Cl chemosensor [169]. The most commonly used analytical method is RP-HPLC with a C18 column and a gradient mobile phase. The UPLC-MS/MS and UHPLC-MS/MS methods have been described for the simultaneous determination of 22 or 5 AEDs, respectively [68,78] in the clinically relevant concentration range of active substances and are suitable for routine TDM in epilepsy polytherapy.

## 11. Lacosamide

Lacosamide (LCM) is a novel inhibitor of sodium channels, developed in 2008 for partial-onset seizures in patients 16 years and older. LCM binds to collapsing response mediator protein-2 (CRMP-2), a phosphoprotein expressed primarily in the nervous system and is involved in neuronal differentiation and control of axonal outgrowth [170]. LCM’s injections are indicated for short term use when oral therapy is not feasible. Its bioavailability is about 100%, and its half-life takes about 13 h. LCM is low bounded to proteins (less than 15%) and catalyzed by CYP2C19 to an inactive metabolite. About 40% of the drug is excreted with urine. Its therapeutic range is 3–10 mg/L [4,125]. Its drug-drug interactions have not been reported except a strong relation with selective serotonin 5-HT3 receptor antagonists or antiretroviral protease inhibitors [171,172]. There are cases also reported of LCM poisonings in attempted suicide [173]. Therefore, the LCM monitoring level is clinically significant for patients with liver or kidney failure, although its pharmacokinetics is predictable for young and adults. LC-DAD, UHPLC-MS/MS and GC-MS detect LCM blood level [174,175,176]. Samples of serum, plasma, whole blood and urine (postmortem and clinical samples) were subjected to alkaline LLE with ethyl acetate, SPE after derivatization (for GC-MS analysis) or protein precipitation with acetonitrile or 60% perchloric acid [174,177]. Nikolaou et al. [177] described a simple GC-MS method with LOQ found to be 0.2 mg/L. LC-DAD technique for determination of LCM in human plasma together with LEV and ZNS was developed. The method was implemented for TDM and individualize the posology of new antiepileptic drugs [109]. Commonly LCM is monitored in clinical practice and forensic toxicology cases.

## 12. Perampanel

Perampanel (PER, 2-(2-oxo-1-phenyl-5-pyridin-2-yl-1,2-dihydropyridin-3-yl-benzonitrate hydrate) is a selective non-competitive α-amino-3-hydroxy-5-methyl-4-isoxazolepropionic acid (AMPA) glutamate receptor antagonist. It has been approved in 2012 and is indicated for patients 12 years of age as well as older with focal and primary generalized tonic-clonic seizures associated with epilepsy [29,44,80,178]. The pharmacokinetics of PER is linear and predictable over the clinically relevant dose range (2–12 mg) [179]. Orally administered PER is rapidly and completely absorbed and is approximately 95% bound to albumin and glycoprotein. PER is extensively metabolized in the liver (>90%), primarily via cytochrome P450 (CYP3A4/5, CYP1A2, CYP2B6) followed by glucuronidation to various pharmacologically inactive metabolites. The half-life is approximately 105 h. PER, like other newer antiepileptic drugs, has a lower risk of drug interactions. Despite its strong protein binding, is not associated with significant interactions with other drugs strongly bound to proteins (warfarin), but the simultaneous administration of CYP3A4 enzymes inducers (e.g., CBZ, PHT, OXC) increases its metabolism and reduces PER plasma concentration even by up to 67% [29,44,45,179]. The current reference range for PER in human plasma is 0.1–1.0 mg/L [4]. Plasma PER and dried spot plasma tests based on HPLC-FLD were described by Mano et al., Franco et al., Mohamed et al. [180,181,182]. The serum PER determination was performed using a capillary [182], LLE [183], and stacking of acetonitrile to pre-concentrate the samples on-line [184]. Fluorescence detection at 290 nm (excitation) and 430 nm (emission) [180,182] and a wavelength between 240–400 nm (excitation) and 495 nm (emission) were used [184]. In the LC-UV method determining PER in a DPS, UV detection was measured at 320 nm [181,185]. All the described methods worked in the therapeutic range and are suitable for TDM of PER [186].

## 13. Rufinamide

Rufinamide (RFM) is a newer anticonvulsant introduced in 2007 on the European market for the treatment of severe epileptic encephalopathy. Since 2008, it is used for Lennox-Gastaut Syndrome in the USA. RFM prolongs the inactive state of voltage-gated sodium channels, thus stabilizing membranes, ultimately blocking the spread of partial seizure activity. The oral bioavailability of RFM administration is about 70–85%, and its half-life takes 4 to 6 h. RFM is medium bound to proteins (40%) and extensively metabolized via enzymatic hydrolysis to inactive product (CGP 47292), excreted with urine. Its therapeutic range is 5–30 mg/L. RFM monitoring level is beneficial because of a good correlation between serum drug level and seizure control to individual needs [4,5,125,187,188]. RFM level is detected by LC coupled with UV or MS detection [189,190,191]. Linear ranges are 0.005–50 mg/L. Samples like a serum, plasma, saliva and DPS, even mouse tissues or postmortem specimen are alkalinized and then LLE extraction with dichloromethane or protein precipitation using acetonitrile or methanol were applied [37,84,192,193]. Gall et al. [191] described the quantification of RFM achieved using LC coupled with tandem mass spectrometry (with very low LOQ 0.005 mg/L) in low volume plasma samples (50 µl). Commonly TDM of RFM is implemented in clinical practice.

## 14. Stiripentol

Stiripentol (STP) was approved as adjunctive therapy for the treatment of severe myoclonic epilepsy in infancy (SMEI) or Dravet Syndrome and can be effective in the reduction of pharmacoresistant seizures and status epilepticus [194,195]. The drug enhances inhibitory GABA-ergic neurotransmission by increasing the activity of GABA_A_ receptors [196]. Its pharmacokinetics is non-linear. Half-life increases with the dose. The drug binds strongly to plasma proteins (99%), exhibits extensive hepatic metabolism by the CYP1A2, CYP2C19, and CYP3A4 isoenzymes, and possesses low bioavailability. As a potent inhibitor of CYP450 complex, the drug increases the plasma concentrations of many other antiepileptic drugs [197,198]. Because of its non-linear kinetics, extensive clearance, and drug-drug interactions, STP should be a good candidate for TDM. Takahashi et al. [199] reported a validated HPLC-FLD method for analysis of STP in plasma of children with Dravet syndrome. The method involves protein precipitation with acetonitrile for sample pretreatment, and chromatographic separation on a C18 column using a mixture of phosphate buffer (pH 2.6) and acetonitrile as a mobile phase. Validation procedure proved high sensitivity of the method with LOQ of 0.05 mg/L. Moreover, the LC-MS/MS method for determination of 22 antiepileptics in postmortem blood, serum and plasma involves an analysis of STP with LOQ of 0.5 mg/L [84].

## 15. Cenobamate

Cenobamate (CNB), the newest antiepileptic medication, marketed as Xcopri (tablets), was approved by the FDA in 2019 to decrease uncontrolled partial-onset seizures in adult patients with epilepsy. The mechanism exerting therapeutic action in patients is not precisely known. However, it is confirmed that inhibits voltage-gated sodium channels and positively modulates γ—aminobutyric acid (γ-GABA) ion channel. CNB administered orally is bioavailable in 88%. The changes of plasma concentration with time (expressed by AUC—area under the curve), increase greater than proportionally in the range of single oral doses 5–750 mg, but the C_max_ increases proportionally with dose. CNB binds in 60%, primarily with a fraction of albumins. The drug possesses a long biological half-life of elimination in the range 50–60 h. The steady-state concentrations in plasma were attained after two weeks. The drug is extensively metabolized by glucuronidation with the use of glucuronosyltransferase-2B7 and oxidation with use isoenzymes mainly CYP2E1, CYP2A6, CYP2B6 as well as CYP2C19 and CYP3A4/5. The unchanged drug, as well as its metabolites, are mainly excreted with urine. CNB represents interactions with other AEDs: Decrease plasma concentrations of LTG, and CBZ, increase levels of PHT, PHB, and N-CLB active metabolite of clobazam. Moreover, CNB decreases plasma levels of CYP2B6 and CYP3A substrates and oral contraceptives, but increase levels of CYP2C19 substrates [1,2,200].

One of the first LC-MS/MS methods was designed for a pharmacokinetics study of CNB concerned determination in rat plasma. Carisbamate was used as the internal standard. Preparation of plasma samples required precipitation of proteins by acetonitrile. The separation was conducted on a reversed-phase C18 column with a mobile phase consisted of 10 mM ammonium formate and acetonitrile (60:40). Detection was performed using a triple quadrupole mass spectrometer by MRM at transitions of *m/z* 268.06 → 198.00 for CNB and *m/z* 216.09 → 198.10 for carisbamate. The calibration curve was linear over a concentration range of 10–5000  ng/mL [201].

An achiral validated LC/MS/MS method for determining CNB in heparinized plasma samples was applied for pharmacokinetic studies. Phenacetin was used as an internal standard. Plasma samples with precipitation of proteins were analyzed. Validated parameters: Accuracy, precision, recovery, LOQ, stability tests confirmed the usefulness of the method for pharmacokinetic studies of CNB administered as tablets [2]. The other LC-MS/MS method was applied for pharmacokinetic studies of CNB in plasma after administering a single capsule formulation with 400 mg containing 50µCi of 14C. The range of the calibration curve was 0.080–40.0 mg/L. The validated liquid scintillation counting method was used for analysis parent drug labelled 14C in whole blood, plasma, urine, and faeces. CNB and eight metabolites were identified in plasma, urine, and faeces using LC-MS/MS coupled with a radio flow detector. The study stated that the penetration of CNB and its metabolites into red blood cells is limited [202].

## 16. Discussion

Methods that meet the validation criteria, including high sensitivity and selectivity, are the basis for TDM, and finally, for effective and safe pharmacotherapy. This statement also applies to AEDs, for which monitoring of drug concentrations becomes the rule nowadays, not only in terms of total concentration (VPA, PHT, CBZ, GBP, LTG, lithium, TPM, LEV) but also of unbound fractions, especially for those drugs with a high (90%) or even higher protein binding rate. Free concentrations of many AEDs including PHT, CBZ, and VPA are in some laboratories routinely determined. An increase in TDM is expected for the most recent AEDs for which the therapeutic concentration ranges have been established. This concerns BRV (0.2–2 mg/L), PER (0.1–1 mg/L), STM (2–10 mg/L) and STP (4–22 mg/L) [4]. For other AEDs: ESL, LTG, OXC, PGB, TPM, and VPA, the reference ranges have been updated or harmonized. It can be expected that free fraction will be determined for other AEDs: CLB, and its active metabolite N-CLB, PER, TGB, retigabine, STP, medicines with high protein binding [203]. There is still a need to develop new, rapid methods that meet the validation criteria. This trend has been observed in the last few years in the bioanalysis of AEDs, where LC-MS/MS is a dominated technique (Appendix A); however, it is costly. Data presented in the review show that near 50% of the LC methods applied the high-resolution detection despite the technical problems with the stability of MS/MS response. The sensitivity connected with the high-resolution LC-MS/MS causes that deuterated analogues of the AEDs are used as effective internal standards even with very small differences in molecular mass as well as retention time [21,22,37,40,42,65,84,119,146,147,149,151,157,176,177]. There is still an increase in the full automatization of analytical determination of AEDs in order to decrease hands-on time as well as consumption of organic solvents, protect the natural environment and improve the analytical process in terms of accuracy, precision, repeatability, and sensitivity. To realize the purpose also different microextraction techniques for AEDs analysis were developed, including ultrasound-assisted emulsification microextraction [47], MEPS [66,75,167], microextraction combined with micro-derivatization-increased detection (MDID) [40], DLLME [59,133]. Automatization was applied for the DBS extraction of VPA, PHB, PHT, CBZ and its active 10,11 epoxide [21]. The effective online SPE coupled with an analytical column of LC-HRMS system was applied for quantification of the first generation of AEDs: PHB, PHT and CBZ and its active metabolite in clinical samples [23]. The interesting idea seems to be online extraction using RACNTs for analysis of CBZ, PHB and PRM, previously used only for extraction cadmium and lead [36].

However, traditional protocols with steps involving sample preparation using traditional LLE and newer SALLE [50], as well as protein precipitation, are frequently applied in the bioanalysis of AEDs, more often than advanced micro-extraction techniques (Appendix A).

Among the presented LC methods, the columns with different length, dimensions and particle sizes based on popular lipophilic chain C-18 are very often applied. Generally, in LC-MS/MS and UHPLC-MS/MS, the columns are characterized by dimensions of 2–3 mm, and smaller than 5 µm particle size (1.7–3.5 µm, Appendix A). The other types of LC columns, like those containing biphenyl phase [88], chiral column Chiracel with normal phase [76], or columns with cyano stationary phase (polar phase) [121] were rarely used in quantification of AEDs. Less popular methods like MEKC gained attention for TDM of PIR [92], capillary electrophoresis for TPM and PRM [122,184]. Worth mentioning is that HPTLC, not so useful in TDM was also developed for determination of OXC and ESL in biological samples [50]. The GC-MS technique loses its importance compared to LC-MS/MS, although it has been developed in recent years to determine TPR [120], GBP [149], LCM [175,177].

Moreover, alternative specimen sampling is proposed employing non-invasive and patient-friendly techniques, including DBS or collection of saliva. These alternative specimens, appropriate for TDM, are especially valuable in specific clinical situations involving pediatric patients or critically ill patients.

## 17. Summary and Conclusions

The increasing development of TDM for AEDs observed in recent years will require more and more perfect, fully automated bioanalytical methods, characterized above all by high sensitivity, selectivity, and short analytical time. It seems that the LC-MS/MS methods can meet these requirements, due to the very high sensitivity and selectivity that go hand in hand with relatively small-milligram doses of AEDs, and the recently observed lowering of therapeutic ranges.

## Figures and Tables

**Figure 1 molecules-25-05083-f001:**
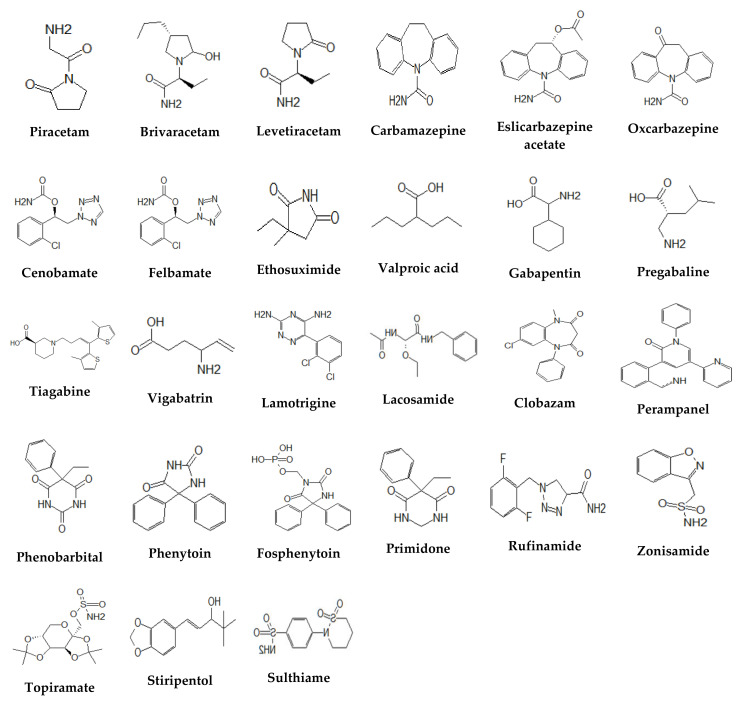
Chemical structures of described AED’s.

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
