# Peer review of "New Methods Used in Pharmacokinetics and Therapeutic Monitoring of the First and Newer Generations of Antiepileptic Drugs (AEDs)"

_molecules, 2020, doi:10.3390/molecules25215083_

Round 1
Reviewer 1 Report
In the abstract, please delete the first sentence.
Please correct grammar errors at line 56.
Lines 64-70: punctuation needs to be corrected.
Throughout the mansucript: check all acronyms and report them in extenso at their first appearance in the text.
The style of Table 1 should be improved.
In the table, it is also reported HPLC-FL and HPLC-FLD: they should refer to HPLC coupled to fluorimetric detection. Please make uniform.
In the table it is not indicated LOD and/or LOQ for any employed tecnique. Also in the text, sometimes LOD and LOQ are missing
I could also suggest to divide the table 1 in two different tables: the first reporting analytical data, and the second the pharmacological properties of AEDs.
The structures of the molecules could be employed
Author Response
We are grateful to the Reviewers for their critical comments. Based on these suggestions, we have made careful modifications to the original manuscript. The point-to-point replies and explanations for all of the revisions are listed below.
In the abstract, please delete the first sentence.
Answer: The first sentence in the abstract was removed.
Please correct grammar errors at line 56.
Answer: Grammar errors were corrected.
Lines 64-70: punctuation needs to be corrected.
Answer: Punctuation was corrected.
Throughout the mansucript: check all acronyms and report them in extenso at their first appearance in the text.
Answer: We checked acronyms and reported them at their first appearance.
The style of Table 1 should be improved.
Answer: The style of Table 1 was improved.
In the table, it is also reported HPLC-FL and HPLC-FLD: they should refer to HPLC coupled to fluorimetric detection. Please make uniform.
Answer: We made HPLC-FLD uniform in the text.
In the table it is not indicated LOD and/or LOQ for any employed tecnique. Also in the text, sometimes LOD and LOQ are missing
Answer: We added missing values of LOQ in Table 2. The LOQ in the main text was mentioned only for the selected methods in order not to repeat the data presented in Table 2.
I could also suggest to divide the table 1 in two different tables: the first reporting analytical data, and the second the pharmacological properties of AEDs.
Answer: Please note that two tables were included in the manuscript. Table 1 presents the pharmacological properties of AEDs and Table 2 reports analytical data.
The structures of the molecules could be employed
Answer: The structures of the molecules were added to the manuscript in the Figure 1.
Reviewer 2 Report
This paper reviews the bioanalytical methods for therapeutic drug monitoring of new antiepileptic drugs. This topic may be of interest to the readers of Molecules. However, various important issues should be addressed. Hence, my main concerns are as follows:
General comments
Figure: Generic structures of each family or representative drugs should be plotted in a figure.
Manuscript organization: The manuscript should be better structured. In my opinion, apart from introduction, new sections could be as follows:
(i) drugs (including drug description with general information on mechanism of action, metabolism, etc.).
(ii) A section focused on sample treatment options according to the different sample matrices.
(iii) A critical revision of analytical methods, with a general discussion of general strategies, and their pros and cons. Currently, this review is a mere summary of abstracts of papers rather that a deeper discussion based on techniques (e.g. chromatographic vs non-chromatographic), separation modes, etc. Besides, the trendiest aspects of the analytical methods should be highlighted (e.g., special chromatographic modes, special column technology, etc.). Additional data on detection wavelengths, MS transitions in MRM mode could be indicated to try to enrich the text.
Tables: I recognize the great efforts of the authors for preparing such a comprehensive revision of antiepileptic drugs, with a lot of references dealing with activity, action mechanism, ADME as well as the analytical methods proposed for their determination, with dozens of references and extensive tables with several pages of length.
Tables are a great work although they should be revised to make uniform all the entries and polish some typos.
Anyway, I have serious concerns on the length of table 2. I do not know if is appropriate to be included in the main manuscript. Perhaps, it could be given as a supplementary material. Instead, I suggest that in the main manuscript authors could focus on the most (few) important drugs and/or representative papers.
Minor comments
UPLC is just a commercial name by Waters. Please use UHPLC instead.
Authors should be consistent with verbal tenses throughout the paper. For instance, in the introduction pass simple, past continuous, present perfect,… are simultaneously used in similar circumstances.
There are some discrepancies in data such as “TDM was introduced in the late 1960s to minimize the toxicity effect” or “The TDM observation was introduced into therapy in the 1940s”, so please clarify these circumstances.
Line 181. Which are the internal standards?
Proton-pomp should be proton-pump.
All acronyms should be defined the first time that appears.
More emphasis and deeper discussion on analytical methods.
Author Response
We are grateful to the Reviewers for their critical comments. Based on these suggestions, we have made careful modifications to the original manuscript. The point-to-point replies and explanations for all of the revisions are listed below.This paper reviews the bioanalytical methods for therapeutic drug monitoring of new antiepileptic drugs. This topic may be of interest to the readers of Molecules. However, various important issues should be addressed. Hence, my main concerns are as follows:
General comments
Figure: Generic structures of each family or representative drugs should be plotted in a figure.
Answer: The structures of the molecules were added to the manuscript in the Figure 1.
Manuscript organization: The manuscript should be better structured. In my opinion, apart from introduction, new sections could be as follows:
(i) drugs (including drug description with general information on mechanism of action, metabolism, etc.).
(ii) A section focused on sample treatment options according to the different sample matrices.
(iii) A critical revision of analytical methods, with a general discussion of general strategies, and their pros and cons. Currently, this review is a mere summary of abstracts of papers rather that a deeper discussion based on techniques (e.g. chromatographic vs non-chromatographic), separation modes, etc. Besides, the trendiest aspects of the analytical methods should be highlighted (e.g., special chromatographic modes, special column technology, etc.). Additional data on detection wavelengths, MS transitions in MRM mode could be indicated to try to enrich the text.
Answer: We dicided not to add new sections as it would require complete reorganization of the manuscript. Our intention was to present clearly the most important analytical aspects of bioanalysis of the each selected AEDs, therefore each drug was described separately. Following your suggestion, the manuscript was supplemented with additional issues including the trendiest aspects of the analytical methods such as special chromatographic modes, special column technology, etc., which were presented in the discussion section. The additional data on detection wavelengths and MS transitions in MRM were added in Table 2.
Tables: I recognize the great efforts of the authors for preparing such a comprehensive revision of antiepileptic drugs, with a lot of references dealing with activity, action mechanism, ADME as well as the analytical methods proposed for their determination, with dozens of references and extensive tables with several pages of length.
Tables are a great work although they should be revised to make uniform all the entries and polish some typos.
Answer: The tables were revised according to your suggestions.
Anyway, I have serious concerns on the length of table 2. I do not know if is appropriate to be included in the main manuscript. Perhaps, it could be given as a supplementary material. Instead, I suggest that in the main manuscript authors could focus on the most (few) important drugs and/or representative papers.
Answer: We decided to include Table 2 in the main text as it would be hard to decide which drugs should be transferred to a supplementary material. The length of the table was left to the Editor’s decision.
Minor comments
UPLC is just a commercial name by Waters. Please use UHPLC instead.
Answer: UPLC was corrected to UHPLC as requested.
Authors should be consistent with verbal tenses throughout the paper. For instance, in the introduction pass simple, past continuous, present perfect,… are simultaneously used in similar circumstances.
Answer: The use of verbal tenses was revised and corrected.
There are some discrepancies in data such as “TDM was introduced in the late 1960s to minimize the toxicity effect” or “The TDM observation was introduced into therapy in the 1940s”, so please clarify these circumstances.
Answer: The sentence “The TDM observation was introduced into therapy in the 1940s” was deleted from the text.
Line 181. Which are the internal standards?
Answer: The names of the internal standards were added.
Proton-pomp should be proton-pump.
Answer: It was corrected as requested.
All acronyms should be defined the first time that appears.
Answer: We checked acronyms and reported them at their first appearance.
More emphasis and deeper discussion on analytical methods.
Answer: The discussion was broadened and supplemented with additional analytical issues.
Round 2
Reviewer 2 Report
I acknowledge that a significant effort has been made to improve the quality of the manuscript.